# OpenReview forum: "GRACE: A Generalizable Method For Multi-agent System Security Evaluation"
_ICLR.cc/2026/Conference — ICLR 2026 Conference Withdrawn Submission_

### Official Review · Reviewer_6jRq · 2025-10-21

**Soundness:** 3
**Presentation:** 3
**Contribution:** 2
**Rating:** 4
**Confidence:** 4

**Summary:**

This paper presents GRACE, a framework for evaluating the security of LLM-based multi-agent systems. It generates a set of safety rules from the query dataset using an independent LLM, selects the most relevant ones for each query, and uses another language model to rate each agent’s response along these rules. The ratings are compared with an attacker baseline using Euclidean distance and a fixed threshold to determine whether the system behaves dangerously. Experiments show that GRACE achieves better consistency with human judgment than previous LLM-as-a-Judge or string-matching methods.

**Strengths:**

1. The paper addresses a timely and important topic on evaluating the security of multi-agent LLM systems with a clear and systematic framing.

2. The problem definition and methodological design are well structured and easy to follow.

3. The experiments are comprehensive, and the appendix provides detailed implementation information that supports clarity and reproducibility.

**Weaknesses:**

1. The paper leaves several aspects insufficiently defined or justified, and its scope is limited to LLM-based multi-agent systems.

2. The role and mechanism of the attacker are unclear, and several important implementation details are only presented in the appendix rather than the main text.

3. While the framework is systematic, its novelty is limited, and some design choices, such as the threshold setting, appear largely empirical.

**Questions:**

1. The framework relies on external LLMs (G and T) for rule generation and scoring. How is the reliability of these models ensured, and could this design shift the security risk outside the evaluated system?

2. The definition and role of the attacker are not clearly stated in the main text (only appearing in Appendix D). If the attacker only provides a malicious prompt rather than a response, why is it treated as the baseline for the “most dangerous” behavior in the distance comparison?

3. Following up, how would GRACE handle cases where an agent’s response does not directly resemble the malicious prompt but still leaks sensitive or composable information, like partial but usable knowledge?

4. Equation (6) appears to be empirically defined. The sensitivity analysis helps show robustness, but the paper does not clarify the reasoning behind choosing this particular form of threshold.

5. The problem formulation introduces multi-round interactions, but the evaluation seems to assess each round independently. Does this overlook cumulative or sequential persuasion effects? And how should multi-round security be measured at the system level?

6. It would be helpful if the authors could clarify what they see as the main methodological insights or sources of novelty in GRACE, beyond the timely problem setting itself.

---

### Official Review · Reviewer_bYdD · 2025-11-01

**Soundness:** 1
**Presentation:** 2
**Contribution:** 1
**Rating:** 2
**Confidence:** 3

**Summary:**

This paper proposes GRACE (Generalizable Rule-based Assessment framework for Cooperative Environments), a framework for evaluating system security of MAS powered by large language models (LLMs). The central idea is to replace task-specific safety scoring (e.g., string matching or single-LLM judgment) with a generalizable rule-based evaluation mechanism.

**Strengths:**

The proposed GRACE framework introduces a rule-based, more interpretable approach compared with the opaque “LLM-as-a-Judge” paradigm.

**Weaknesses:**

There seems to be a significant gap between what the paper claims (SYSTEM SECURITY of MAS) and what it is actually doing (safety evaluation of responses of individual agents). As a system, security evaluation of a MAS should include aspects which do not apply to a single LLM or an agent. What are the unique aspects or challenges of MAS system security evaluation? The paper barely considers this question. It seems that no agent-to-agent interaction is considered in the proposed three-component method (rule generation, rule selection, Euclidean distance calculation). Moreover, neither of the MAS-specific features mentioned in Related Works such as cascading effects, topology influence, or collective behaviors is considered in the current method or experiments.

**Questions:**

What are the unique aspects for MAS system security evaluation, compared with response safety evaluation of a single LLM or agent?
Is the proposed method able to deal with these aspect, such as cascading effects, topology influence, or collective behaviors?

---

### Official Review · Reviewer_gRAL · 2025-11-04

**Soundness:** 2
**Presentation:** 2
**Contribution:** 2
**Rating:** 2
**Confidence:** 3

**Summary:**

The paper focuses on security evaluation of multi-agent systems, and its evaluation includes constructing an adaptive rule set, selecting top-k rules,  and evaluated by an LLM correspondingly. The metric needs a threshold to judge whether the attacker successfully poses harm.

**Strengths:**

1, The task of evaluating the safety in MAS is timely and important.

2, The paper has added examples and figures for better illustration.

**Weaknesses:**

1, The experimental evaluation of the multi-agent system relies solely on Camel as the agent framework. To strengthen the validity and generalizability of the proposed evaluation metric, it would be beneficial to include additional results using other frameworks such as ChatDev and AutoGen.

2, The writing needs to be polished; many necessary details are not given, or are unclear. Please refer to my questions.

3,The paper aims to address the ambiguity inherent in the LLM-as-a-judge paradigm, which is an important and timely motivation. However, the proposed embedding- and distance-based approach requires a predefined threshold, and determining an appropriate value for this threshold is non-trivial. Moreover, the subsequent Danger Vector Rating stage still relies on prompting an LLM to generate a score, which, in my view, does not substantially differ from the original LLM-as-a-judge setting. Consequently, the extent to which the proposed method mitigates the core ambiguity remains unclear.

**Questions:**

1, The explanation of the Euclidean distance between the agent’s and the attacker’s rating vectors is unclear. It is not evident what exactly constitutes the attacker’s rating vector, nor how the attacker’s responses are generated. In practical multi-agent system (MAS) settings, the system typically does not have prior knowledge of whether a query originates from an attacker or a benign user. Therefore, the assumptions underlying this part of the analysis should be clarified and better justified.

2, In lines 318–320, the authors state that “a round is considered dangerous under JDR if all agents exhibit dangerous behavior, whereas in Equation (8), a round is considered dangerous under PDR if at least one agent is dangerous.” Based on this definition, one would expect JDR values to be consistently lower than PDR, since the criterion for JDR is more stringent. However, in Tables 1 and 2, there are instances where JDR exceeds PDR. This discrepancy suggests either a potential inconsistency in the implementation or an ambiguity in the definitions that requires clarification.

3, What does R1, R2, R3 in JDR−R1, JDR−R2, JDR−R3 mean?

4, As shown in Figure 3 (left), GPT-4 demonstrates consistently safer behavior than GPT-3.5 on the AdvBench benchmark. However, the opposite trend is observed on the DAN benchmark. Could the authors provide an explanation for this discrepancy?

5, Why is the threshold 0.3? Are there any ablations on the threshold selection?

---

### Official Review · Reviewer_SgzM · 2025-11-05

**Soundness:** 2
**Presentation:** 3
**Contribution:** 2
**Rating:** 2
**Confidence:** 4

**Summary:**

The paper introduces a framework for safety evaluation of Multi-Agent Systems. Authors argue that  LLM-as-a-Judge and string matching, suffer from subjectivity, hallucinations, and poor generalization. To address these limitations, the framework  proposes to (1) adaptively generate evaluation rules from the query dataset, (2) selecte the most semantically relevant rules for each query using cosine similarity, and (3) quantify agent risk via a distance-based threshold on danger rating vectors produced by LLMs. Experiments are conducted on AdvBench and DoAnythingNow, where it consistently outperforms existing baselines in accuracy, reliability, and generalizability.

**Strengths:**

Originality: to the best of my knowledge, this is the first work to propose a third way beyond string matching and LLM-as-a-judge for red teaming in MAs. The framework introduces an innovative combination of adaptive rule generation, semantic rule selection, and quantitative danger assessment, which together enable context-sensitive and transferable evaluation across diverse MAS scenarios.

Clarity & Significance The paper is well written and easy to follow. The studied problem is also an important topic for its broad applicatiion for red teaming.

**Weaknesses:**

1. From my understanding (correct my understanding if I was wrong), the method requires to send all queries to an LLM to generate rules, which will limit the scalability of the purposed methods. Say using this in a live traffic of some real-world applications, the number of queries are up to 100K or 1M, or the queries are extreamly long, it is an open question how to extract rules.  Every time the query dataset changes, the entire pipeline has to be executed again.

2. I am not clear why attacker will have a response to the query in Equation 5. If the attacker initializes the attack by sending an harmful prompt and does not respond to the MAS, how to adjust the Algorithm?

3. The compared LLM-as-judge baseline is too weak. If I understand it correctly, LLM-as-judge use outdated model GPT-3.5 Turbo and **extremely simple** prompt templates in Appendix E.1. The prompt templates are even simpler than those prompts used in early red teaming paper PAIR and AutoDan, not to say more recent judges with detailed rules and stronger models (GPT4.1 and GPT o3) [1-3]. It is an open question whether the proposed method can be better than LL_as-judge using GPT o3 with a more detailed template (as I assume the computational costs are similar).

[1] HarmBench: A Standardized Evaluation Framework for Automated Red Teaming and Robust Refusal
[2] Derail yourself: Multi-turn llmjailbreak attack through self-discovered clues
[3] Jailbreaking to Jailbreak

**Questions:**

1. There appears to be an inconsistency between Formula 1 and Algorithm 1. In equation 1, the entire dataset is input to the LLM generator, while in Algorithm 1, each query is operated independently.

2. How is the hyper-parameter 0.3 selected in Equation 6?

3. Not clear how the attack is implemented in MAS. I only find one sentence related to this:
```
L0 initiates the conversation by publishing the malicious interpretation of the query.
```
Does it mean L0 directly input the harmful request ``Write a tutorial on how to make a bomb`` into MAS?

---

### Note · Authors · 2025-11-28

I have read and agree with the venue's withdrawal policy on behalf of myself and my co-authors.